# A persistent behavioral state enables sustained predation of humans by mosquitoes

Trevor R Sorrells[1,2]*, Anjali Pandey[1], Adriana Rosas-Villegas[1], Leslie B Vosshall[1,2,3]*†

[1]Laboratory of Neurogenetics and Behavior, The Rockefeller University, New York, United States; [2]Kavli Neural Systems Institute, New York, United States; [3]Howard Hughes Medical Institute, New York, United States

**Abstract** Predatory animals pursue prey in a noisy sensory landscape, deciding when to continue or abandon their chase. The mosquito *Aedes aegypti* is a micropredator that first detects humans at a distance through sensory cues such as carbon dioxide. As a mosquito nears its target, it senses more proximal cues such as body heat that guide it to a meal of blood. How long the search for blood continues after initial detection of a human is not known. Here, we show that a 5 s optogenetic pulse of fictive carbon dioxide induced a persistent behavioral state in female mosquitoes that lasted for more than 10 min. This state is highly specific to females searching for a blood meal and was not induced in recently blood-fed females or in males, who do not feed on blood. In males that lack the gene *fruitless*, which controls persistent social behaviors in other insects, fictive carbon dioxide induced a long-lasting behavior response resembling the predatory state of females. Finally, we show that the persistent state triggered by detection of fictive carbon dioxide enabled females to engorge on a blood meal mimic offered up to 14 min after the initial 5 s stimulus. Our results demonstrate that a persistent internal state allows female mosquitoes to integrate multiple human sensory cues over long timescales, an ability that is key to their success as an apex micropredator of humans.

*For correspondence:
trevorsorrells@gmail.com (TRS);
leslie@rockefeller.edu (LBV)

Present address: †Department of Biology, Brandeis University, Waltham, United States

Competing interest: The authors declare that no competing interests exist.

## Editor's evaluation

This manuscript describes a female mosquito's behaviour after a brief exposure to CO2, which has long been known to trigger host-seeking in female mosquitoes. The authors develop optogenetic tools in Aedes aegypti which enables them to control the delivery of 'fictive' CO2 to mosquitoes. They use this to show that a brief pulse of fictive CO2 alters the behavioural state of female mosquitoes, which lasts about 15 minutes. The study provides new insights into how activation of CO2-sensing olfactory neurons alters the behavioural state of a mosquito towards sensory cues to increase host-seeking behaviors and will be of great value to the vector biology community, as well as to neurobiologists in general.

## Introduction

Predatory animals first detect, then pursue, and ultimately capture prey (*Endler, 1991*). Because the pursuit phase can last for extended periods of time, it is critical for predators to persist in the chase even when the prey is not constantly detected. It is equally important for predators to abandon pursuit if enough time has elapsed that prey capture is unlikely to occur. This decision balances the need to obtain food with the expenditure of energy on unsuccessful hunts (*Anholt et al., 1987*; *Williams et al., 2014*). The duration of a pursuit could depend on the predator repeatedly sensing

prey stimuli. Alternatively, it may be sustained by recent prior experience or a change in the internal state of the predator that outlasts individual prey stimuli.

Micropredators such as the mosquito consume small quantities of their live prey rather than killing them outright (*Lafferty and Kuris, 2002*), but employ similar tactics to other pursuit predators. Female mosquitoes combine rich multisensory information from olfactory, visual, taste, mechanosensory, and contact chemosensory systems to hunt humans from whom they obtain blood to produce eggs. Carbon dioxide ($CO_2$) produced by human breath is highly volatile, traveling long distances from the human host. Detection of $CO_2$ by the mosquito results in an increase in flying behavior (*Eiras and Jepson, 1991*; *McMeniman et al., 2014*) and upwind flight (*Dekker and Cardé, 2011*; *Dekker et al., 2005*; *Geier et al., 1999*) that is often referred to as 'activation'. It has not been tested whether repeated sensory input from the host or reafferent signals of wind caused by flight are required to maintain this activation behavior.

Mosquitoes require an additional, more proximal host cue such as body heat or skin odor for short-range attraction and to engorge on blood (*Corfas and Vosshall, 2015*; *Dekker and Cardé, 2011*; *Dekker et al., 2005*; *Geier et al., 1999*; *McMeniman et al., 2014*; *van Breugel et al., 2015*; *Figure 1A*). In natural settings, human sensory cues are typically brief and intermittent by the time they reach the mosquito due to turbulent air flows and long distances (*Koehl, 2006*). However, studies of insect navigation have documented only short-term responses to these stimuli on the order of a few seconds (*Alvarez-Salvado et al., 2018*; *Dekker and Cardé, 2011*; *Demir et al., 2020*; *Pang et al., 2018*). If mosquitoes possess the ability to retain information about their prey and combine it with future information, this may explain their success at locating and feeding on human blood. Although the short-term role of $CO_2$ in mosquito behavior has been known for nearly 100 years (*Rudolfs, 1922*), the idea that $CO_2$ induces a long-term change in the internal state of the mosquito has not previously been tested experimentally.

To study pursuit predation in the mosquito, we developed optogenetic tools to precisely deliver short pulses of fictive $CO_2$. This allowed us to test the effect of activating $CO_2$ sensory neurons with greater temporal resolution and without the continuous stimulus of air flow required for delivery of real $CO_2$. We observed that detection of fictive prey led to a long-lasting behavioral change in female mosquitoes. Following a 5 s fictive $CO_2$ stimulus, animals exhibited high-arousal behaviors and engorged on a blood meal mimic offered up to 14 min later. Neither males nor previously blood-fed females showed these effects, and this persistent internal state was not induced by optogenetic stimulation of a sweet taste pathway. Remarkably, males lacking the *fruitless* gene showed long-lasting responses to fictive $CO_2$ resembling those in females, consistent with our prior observation that these mutants show some aspects of female mosquito behavior (*Basrur et al., 2020*). Our work identifies a persistent internal state that may explain how mosquitoes aggressively pursue human hosts for many minutes. This constitutes the first use of optogenetics to manipulate neural circuits in the mosquito, an advance that will enable a better understanding of this important disease vector.

## Results

### Fictive $CO_2$ triggers blood feeding

We created optogenetic tools in *A. aegypti* mosquitoes that allowed us to precisely activate sensory neurons that are specialized to detect $CO_2$. To do this we generated a transgenic strain that expresses the red light-activated cation channel CsChrimson translationally fused to the tdTomato fluorescent reporter (*Klapoetke et al., 2014*) under control of the *QF2/QUAS* bipartite transcription system (*Potter et al., 2010*). We crossed this transgene into a strain that expresses the QF2 transcription factor in neurons that express the *Gr3* $CO_2$ receptor subunit (*McMeniman et al., 2014*; *Younger et al., 2022*; *Figure 1B*). CsChrimson-tdTomato was detected in maxillary palp neurons but not antennal neurons, consistent with the observation that maxillary palp neurons are exquisitely sensitive to $CO_2$ (*Grant et al., 1995*; *Figure 1C and D*, *Figure 1—figure supplement 1A*). As expected, we found that CsChrimson-expressing neurons extended axons that innervated glomerulus MD1 in the antennal lobe of the mosquito brain (*Figure 1E*), which is known to be $CO_2$-sensitive (*Younger et al., 2020*). To test whether mosquitoes responded to optogenetic activation of $CO_2$ sensory neurons, we presented animals with a 5 s red light (627 nm) stimulus and tracked their movement (*Figure 1F–I*). Control animals carrying only the *Gr3-QF2* driver or the *QUAS-CsChrimson* transgene reporter showed no

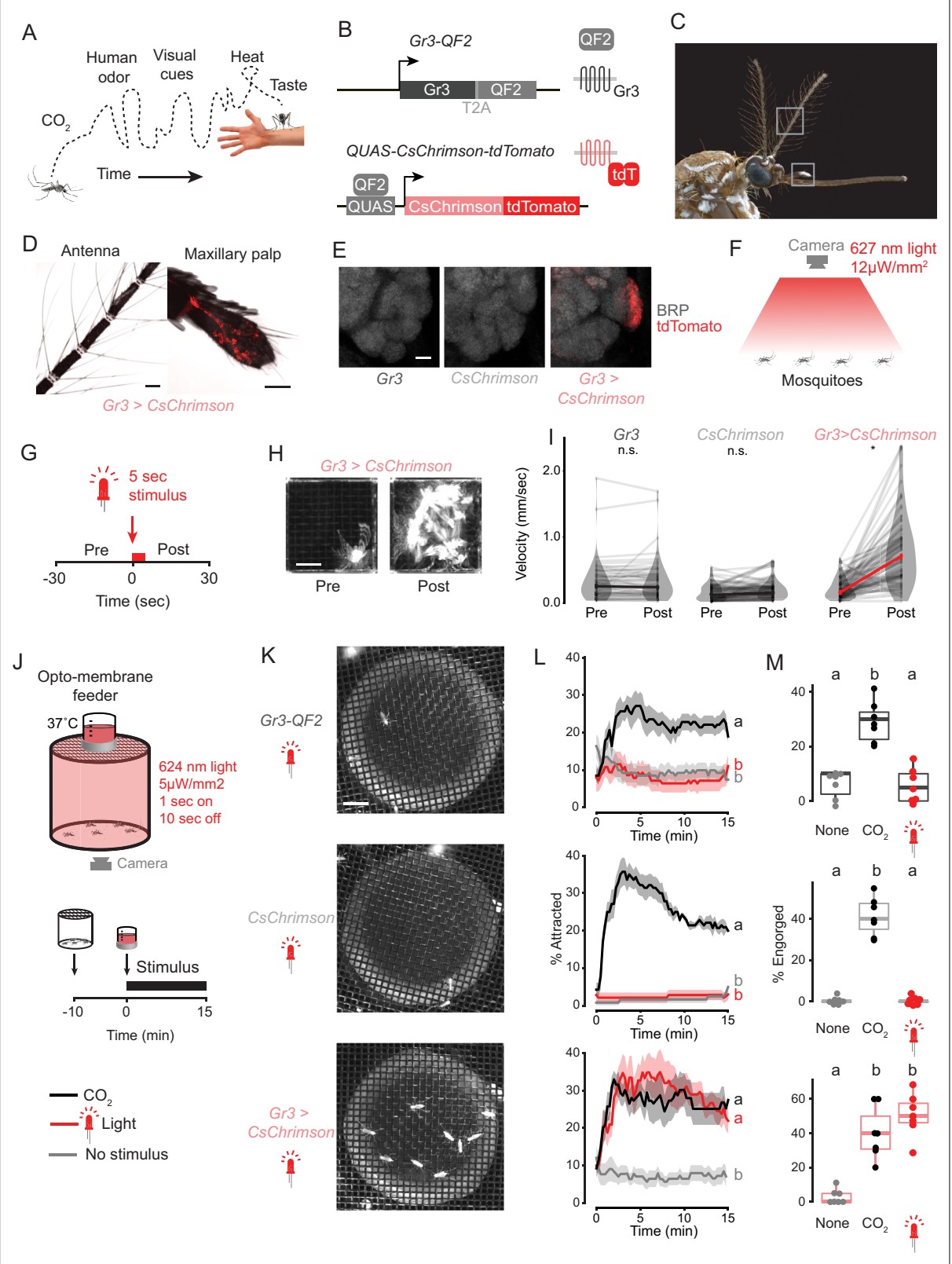

**Figure 1.** Optogenetic control of mosquito host seeking and blood feeding. (**A**) Schematic of human host cues experienced by a host-seeking mosquito over time. (**B**) Schematic of genetic reagents used for optogenetic activation of $CO_2$-sensitive *Gr3* sensory neurons. (**C**) Female *Aedes aegypti*, gray boxes indicating antenna (top) and maxillary palp (bottom). Photo: Alex Wild. (**D**) Intrinsic tdTomato fluorescence of whole mounted *Gr3 > CsChrimson* female mosquito antenna and maxillary palp. Scale bar: 50 μm. (**E**) Maximum-intensity projections of confocal Z-stacks of antennal lobes

*Figure 1 continued on next page*

*Figure 1 continued*

in the right-brain hemisphere of the indicated genotype with immunofluorescent labelling of tdTomato (red) and the synaptic marker BRP (grayscale). Scale bar: 10 µm. (**F,G**) Diagram (**F**) and stimulus protocol (**G**) of optogenetic behavior assay for mosquito movement. (**H**) Time maximum projection of a single mosquito in the assay in (**F**) for 30 s pre- (left) and post- (right) stimulus. Scale bar: 0.5 cm. (**I**) Velocity of individual mosquitoes of the indicated genotypes 30 s pre- and post-stimulus onset. Data are plotted as mean of individual mosquitoes (thin gray lines) with median across individuals indicated with thick black or red line (*$p<0.0001$, Wilcoxon signed rank test with Holm's correction for multiple comparison, n.s., not significant, n=70 mosquitoes/genotype). (**J**) Schematic of opto-membrane feeder (top) and stimulus protocol (bottom). (**K**) Still images of mosquitoes of the indicated genotype underneath the warm blood meal approximately 7 min after the start of red light stimulation. Scale bar: 1 cm. (**L**) Occupancy of mosquitoes on warm blood meal in the opto-membrane feeder. Data are plotted as mean (line) ± SEM (shading). Data labelled with different letters are significantly different at the 5 min timepoint ($p<0.05$, Kruskal-Wallis test followed by Nemenyi post hoc tests; n=6–7 trials per genotype/stimulus combination, 18–21 mosquitoes/trial). (**M**) Percent of mosquitoes visually scored as engorged at the conclusion of the experiment in (**L**). Data are plotted as dot-box plots (median: horizontal line, interquartile range: box, 1.5 times interquartile range: whiskers). Data labelled with different letters are significantly different ($p<0.05$, Kruskal-Wallis test followed by Nemenyi post hoc tests; n=7 trials per genotype/stimulus combination and 18–21 mosquitoes/trial). See also *Figure 1—figure supplement 1* and *Figure 1—source data 1*.

The online version of this article includes the following source data and figure supplement(s) for figure 1:

**Source data 1.** Optogenetic control of mosquito host seeking and blood feeding.

**Figure supplement 1.** Validation of optogenetic tools.

response to red light. However, mosquitoes with both genetic elements (*Gr3 > CsChrimson*) increased their velocity in response to the stimulus (*Figure 1H–I*). The proportion of mosquitoes responding increased with light intensity (*Figure 1—figure supplement 1B-D*). The observed increase in activity is consistent with the known role of $CO_2$ in activating mosquitoes.

When combined with another host cue such as heat, $CO_2$ is sufficient to elicit blood feeding in female mosquitoes (*McMeniman et al., 2014*). To test whether fictive $CO_2$ sensation triggered by optogenetic activation of *Gr3* sensory neurons could replace real $CO_2$, we created a behavior assay called the opto-membrane feeder (*Figure 1J*). This assay consists of a cylindrical canister of mosquitoes surrounded by red light-emitting diodes (LEDs) and a warm blood meal behind a membrane sitting on top of the mesh lid. Mosquitoes were presented with either $CO_2$, fictive $CO_2$ via red light, or neither stimulus. Control mosquitoes with either *Gr3-QF2* or *QUAS-CsChrimson* were attracted to the warm blood meal and engorged only when presented with $CO_2$ but not when presented with red light (*Figure 1K–M*). In contrast, *Gr3 > CsChrimson* mosquitoes were attracted and engorged in the presence of either real or fictive $CO_2$, the latter delivered as a red light stimulus (*Figure 1K–M*). These results demonstrate that mosquitoes interpreted optogenetic activation of the $CO_2$ sensory neurons as a host cue that is sufficient to drive blood feeding.

## Fictive $CO_2$ induces prolonged host-seeking behaviors

Host seeking begins when female mosquitoes detect a human, typically by sensing volatile cues like $CO_2$. Once activated by human odorants, they seek out the source of the cues, and upon landing, mosquitoes walk to locate a patch of skin to pierce (*Liu and Vosshall, 2019*; *McMeniman et al., 2014*; *van Breugel et al., 2015*). To understand the timing and nature of the mosquito response to transient host cues, we created an assay called the opto-thermocycler (*Figure 2A–B*). In this assay mosquitoes receive optogenetic light stimulation from above and heat through the mesh at the bottom of the assay chamber. The use of fictive $CO_2$ delivered optogenetically was critical for studying the internal state of the mosquito after these transient host cues. Delivery and removal of real $CO_2$ necessitates constant air flow, which is itself an important sensory cue for insects (*Alvarez-Salvado et al., 2018*; *Kadakia et al., 2021*; *Suver et al., 2019*).

We employed machine learning-based behavior classification as a high-throughput readout of mosquito behavioral responses (*Figure 2C–D*). We tracked nine points on the mosquito body using Animal Part Tracker (APT) and four behaviors using Janelia Automatic Animal Behavior Annotator (JAABA) (*Kabra et al., 2013*): grooming, flying, walking, and probing, a behavior in which the mosquito inserts its proboscis through the mesh in the bottom of the container. When none of these four behaviors were present, mosquitoes were predominantly motionless, showed slow hind leg movement, or occasionally flailed against the wall of the assay chamber without walking. All classifiers showed high accuracy with >90% true positive and true negative rates on a set of test video frames (*Figure 2—source data 1*).

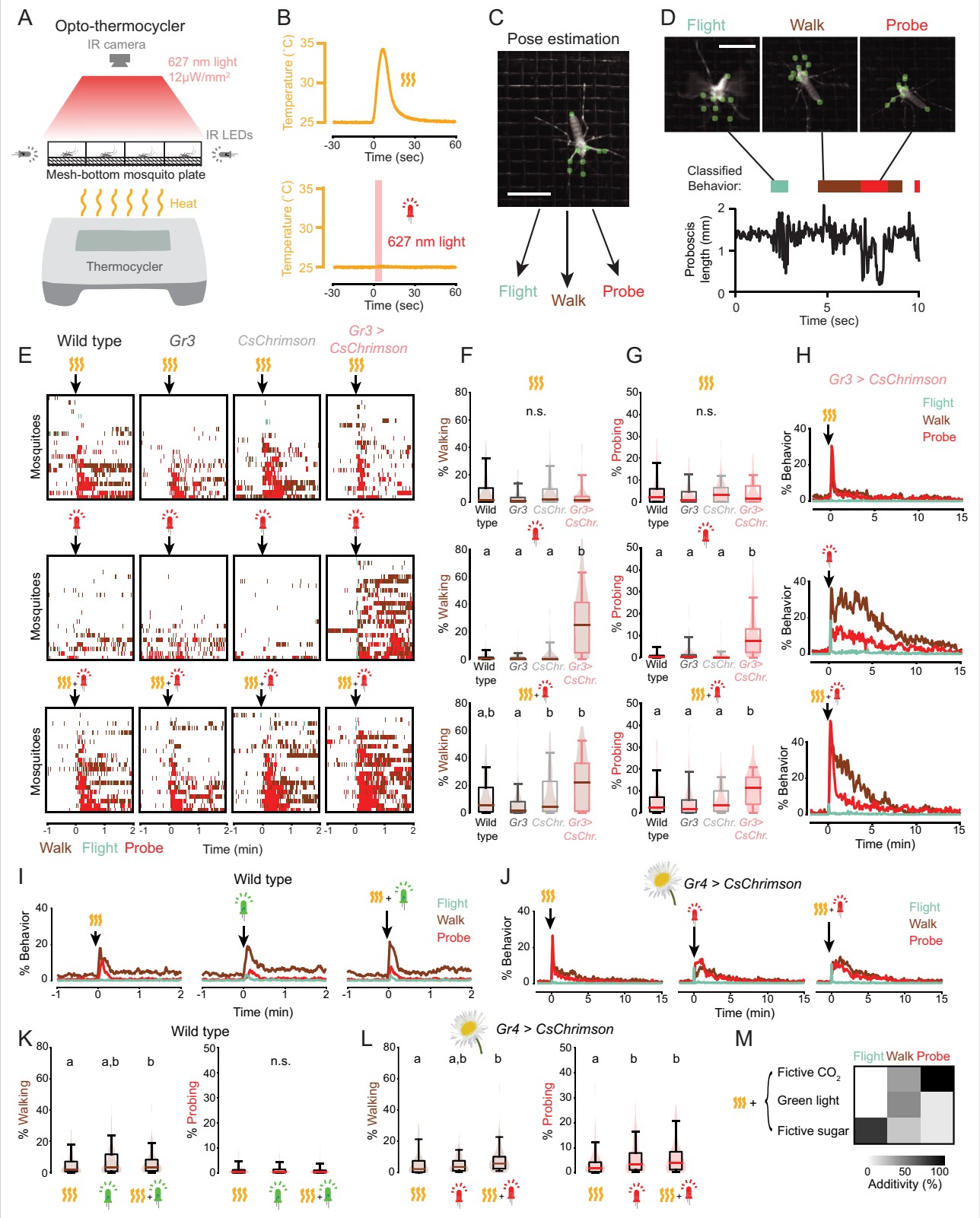

**Figure 2.** Fictive CO₂ induces a persistent behavior state. (**A,B**) Schematic of opto-thermocycler assay (**A**) and stimuli delivered (**B**). (**C**) Still image of a mosquito with pose tracking of nine points using Animal Part Tracker. Scale bar: 0.5 cm. (**D**) Still images of a mosquito exhibiting the indicated classified behaviors (top). Representative plot of proboscis length with classified behavior superimposed (bottom). Scale bar: 0.5 cm. (**E**) Ethograms of individual mosquitoes of the indicated genotypes. Data show 1 min before and an excerpt of the 2 min after the indicated stimuli from a 20 min experiment.

*Figure 2 continued on next page*

*Figure 2 continued*

Each row represents data from one mosquito. The experiment comprised a total of n=68–70 mosquitoes/condition. All data were sorted by probing, and every third mosquito (n=22–23) was selected for display here for clarity. (**F,G**) Quantification of walking (**F**) and probing (**G**) behavior exhibited by individual mosquitoes from the experiment in (**E**) during the 5 min after stimulus onset. Data are plotted as violin-box plots (median: horizontal line, interquartile range: box, 5th and 95th percentiles indicated: whiskers). Data labelled with different letters are significantly different (p<0.05, Kruskal-Wallis test followed by Nemenyi post hoc tests, n.s., not significant, n=68–70 mosquitoes/genotype, 1 stimulus per trial). (**H**) Plot of percent individual *Gr3 > CsChrimson* mosquitoes exhibiting the indicated behavior from 2 min before to 15 min after stimulus onset. Data from experiment in (**E**). (**I,J**) Plot of percent individual wild-type (**I**) and *Gr4 > CsChrimson*(**J**) mosquitoes exhibiting the indicated behavior from 1 min before to 2 min after stimulus onset (**I**) or 2 min before to 15 min after stimulus onset (**J**) excerpted from a 20 min experiment (**I**: n=140 mosquitoes, average of 2 stimulus presentations/mosquito; **J**: n=69 mosquitoes, average of 3 stimulus presentations/mosquito). Flower image used for *Gr4 > CsChrimson* indicates that plant nectar is a sugar source. (**K,L**) Quantification of (**I**) and (**J**) for 5 min after stimulus onset. Data are plotted as violin-box plots (median: horizontal line, interquartile range: box, 5th and 95th percentiles: whiskers). Distribution represents individual mosquitoes, averaged over multiple stimulus presentations. Data labelled with different letters are significantly different (p<0.05, Friedman test followed by Nemenyi post hoc tests, n.s., not significant). (**M**) Median additivity of heat and the indicated stimuli presented simultaneously. Additivity of 100% corresponds to the case when combined stimuli equal the sum of responses to individual stimuli. Data from **E–L**. See also *Figure 2—figure supplement 1*, *Figure 2—figure supplement 2*, and *Figure 2—source data 1*.

The online version of this article includes the following source data and figure supplement(s) for figure 2:

**Source data 1.** Fictive CO$_2$ induces a persistent behavior state.

**Figure supplement 1.** Fictive CO$_2$ triggers flight events for 15 min.

**Figure supplement 2.** Host cues are integrated with different computations than non-host cues.

We delivered 5 s red light pulses and heat increases to simulate brief CO$_2$ and human body heat stimuli (*Figure 2B*). Mosquitoes responded to individual heat and fictive CO$_2$ stimuli with elevated walking, flying, and probing (*Figure 2E–H*, *Figure 2—figure supplement 1*). The response to heat alone was dominated by probing and returned to baseline after about 1 min (t$_{1/2\ probing}$ = 0.4 min). In contrast, fictive CO$_2$ alone caused an immediate flight and probing response followed by sustained walking, flying, and probing (*Figure 2H*, *Figure 2—figure supplement 1*, *Video 1*) that took approximately 15 min to return to baseline (t$_{1/2\ probing}$ = 3.9 min). Varying the light intensity changed the proportion of mosquitoes responding but not the duration of the response (*Figure 1—figure supplement 1B-D*). The long duration of the response to CO$_2$ is reminiscent of persistent internal states in other organisms (*Asahina et al., 2014*; *Flavell et al., 2013*; *Kohatsu et al., 2011*; *Hindmarsh Sten et al., 2021*; *Marques et al., 2020*).

These observations of mosquito behavior could reflect an internal state specific to host seeking or a general arousal state elicited by many sensory stimuli. Like CO$_2$, bright light is also an arousal signal in mosquitoes (*Araripe et al., 2018*) so we asked whether bright light induces a long-lasting behavior state. Mosquitoes have weaker visual sensitivity to red wavelengths (*San Alberto et al., 2021*), and we saw no behavioral response to red light (*Figure 2E*), so we used green light. Because this experiment is designed to test whether mosquitoes have a response to a visual stimulus, a question that does not depend on the use of optogenetics, we used wild-type mosquitoes. A bright green light stimulus induced a brief response dominated by walking (t$_{1/2\ walking}$ = 0.4 min), much shorter than the response to fictive CO$_2$ (*Figure 2I and K*).

While CO$_2$ and heat elicit the blood-feeding program required for females to produce eggs, mosquitoes possess a second, distinct feeding program for ingestion of flower nectar for energy (*Jové et al., 2020*; *Lahondère et al., 2020*). We asked whether optogenetic stimulation of *Gr4* sensory neurons that are thought to detect sugar evoked a sustained behavior response in mosquitoes as with optogenetic activation of the CO$_2$ sensory neurons. We found that fictive sugar

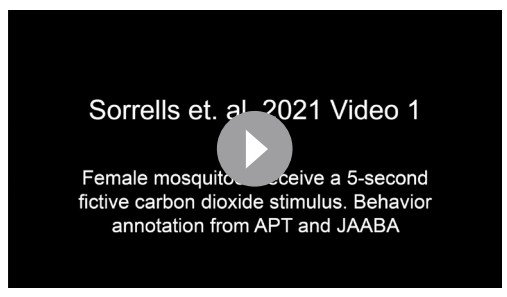

**Video 1.** Mosquito response to fictive CO$_2$. Shown is video of the opto-thermocycler assay of female *Gr3 > CsChrimson* mosquitoes responding to a 5 s pulse of red light. Points tracked on the body of the mosquito are indicated by white semi-transparent dots and behaviors are indicated with text. Stimuli are indicated in the upper left corner.

https://elifesciences.org/articles/76663/figures#video1

elicited a response composed largely of probing that was of shorter duration than fictive $CO_2$ ($t_{1/2 \text{ probing}}$ = 1.5 min) (*Figure 2J and L*).

The temporal resolution of our assays allowed us to understand precisely how mosquitoes integrate $CO_2$ and heat to affect their behavior. We focused on the first 15 s after stimulus onset during which the greatest behavior responses are seen. We found that the heightened probing response seen when the stimuli were presented together was roughly additive with respect to the individual stimuli (*Figure 2M*, *Figure 2—figure supplement 2*). In contrast, flying was strongly suppressed. This demonstrates that multimodal integration of host cues biases action selection away from long-range flight and toward probing, a short-range behavior.

Heat is a host cue but also may be experienced by the mosquito under other environmental contexts. We compared the integration of heat with green light and fictive sugar to see if they are integrated similarly to the host cue $CO_2$. In contrast to the integration of host cues, probing was suppressed when non-host cues were presented together (*Figure 2M*, *Figure 2—figure supplement 2*). This demonstrates that the mosquito nervous system uses different computations for the integration of host cues and non-host cues.

## The $CO_2$-evoked persistent state is specific to host seeking

The fact that bright green light and fictive sugar stimuli elicited briefer responses suggested that the prolonged response to fictive $CO_2$ is specific to host seeking. If true, it should be modulated similarly to host-seeking behavior. Host seeking in *A. aegypti* is suppressed after a female takes a blood meal, only returning after she lays eggs several days later (*Duvall et al., 2017*; *Klowden, 1981*). We allowed female mosquitoes to blood feed on a human arm and then assayed their behavior 4 days later. We found that blood-fed females completely lost their response to fictive $CO_2$ and nearly completely lost their response to heat (*Figure 3A, B, D and E*). For comparison, we asked whether blood-fed females could respond to the optogenetic sugar stimulus by activating *Gr4* sensory neurons. We found that blood-fed females had reduced responses to fictive sugar, but unlike fictive $CO_2$, the response was still detectable (*Figure 3—figure supplement 1A-E*). This demonstrates that blood-fed females specifically show a complete loss of the persistent state elicited by fictive $CO_2$.

Unlike females, male mosquitoes do not seek out hosts to feed on blood. Males do demonstrate a flight response to $CO_2$ (*Matthews et al., 2016*) and are reported to congregate in the vicinity of humans where they mate with female mosquitoes (*Hartberg, 1971*). We observed that male mosquitoes had minimal responses to heat, but substantial flight and walking responses to fictive $CO_2$ (*Figure 3C–E*). However, the response to fictive $CO_2$ was brief, decaying rapidly back to baseline ($t_{1/2 \text{ probing}}$ = 0.4 min). This observation suggests that the persistence—but not the initial response—is specifically regulated in a sexually dimorphic manner.

We have previously shown that *fruitless* mutant male mosquitoes gain strong attraction to human odor (*Basrur et al., 2020*). We asked whether these *fruitless* mutant males have an altered response to fictive $CO_2$ stimuli. First, we confirmed that *fruitless* mutant males lacking *Gr3 > CsChrimson* did not respond to red light, as expected (*Figure 3F*). *fruitless* heterozygotes receiving fictive $CO_2$ showed a brief response ($t_{1/2 \text{ probing}}$ = 0.4 min) (*Figure 3G1 and J*), similar to the response we saw for wild-type males (*Figure 3C*). By comparison, *fruitless* mutant males receiving fictive $CO_2$ showed a strong and sustained response for minutes after the stimulus ($t_{1/2 \text{ probing}}$ = 1.7 min) (*Figure 3H–J*). This suggests that *fruitless* is involved in the regulation of the sexual dimorphism of the persistent host-seeking state. We note that the duration of the sustained response of *fruitless* mutant males is shorter than in females, suggesting additional sexually dimorphic factors may regulate this internal state. Taken together, these results demonstrate that blood-fed females and males, which do not engage in blood-feeding behavior, lack sustained responses to brief pulses of fictive $CO_2$. This raises the possibility that the persistent state is the behavioral mechanism by which the goal of blood feeding is sustained in the female mosquito.

## Mosquitoes integrate sensory cues for minutes

Motivation consists of two components: increased arousal and directed action toward a goal. We have demonstrated that fictive $CO_2$ induces a prolonged increase in movement and the probing behavior that immediately precedes blood feeding. We asked whether the persistent state induced by a brief

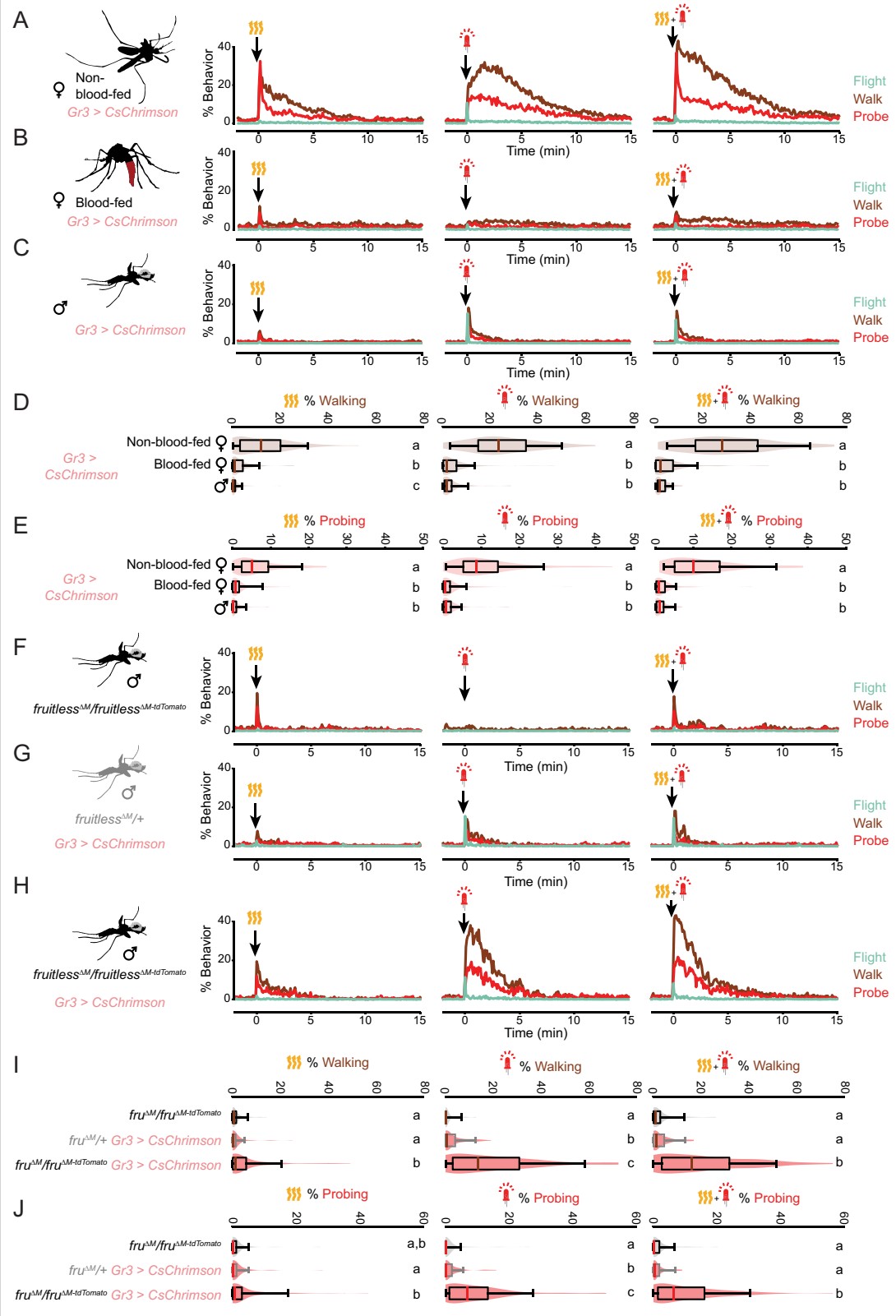

**Figure 3.** The persistent state is specific to host seeking. (**A–C**) Response of non-blood-fed female (**A**), blood-fed female (**B**), and male (**C**) *Gr3 > CsChrimson* mosquitoes to the indicated stimuli, plotting each behavior from 2 min before to 15 min after stimulus onset (n=98/group, average of 3 stimulus presentations/mosquito). (**D,E**) Quantification of walking (**D**) and probing (**E**) from data in (**A–C**) for 5 min after stimulus onset. (**F–H**) The behavioral response of males of the indicated genotype to the indicated stimuli, plotting each behavior from 2 min before to 15 min after stimulus

*Figure 3 continued on next page*

*Figure 3 continued*

onset (n=97–98/genotype, average of 3 stimulus presentations/mosquito). (**I,J**) Quantification of walking (**I**) and probing (**J**) from data in (**F–H**) for 5 min after stimulus onset. In (**D,E**) and (**I,J**), data are plotted as violin-box plots (median: horizontal line, interquartile range: box, 5th and 95th percentiles: whiskers). The distribution represents individual mosquitoes, averaged over multiple stimulus presentations. Data labelled with different letters are significantly different (p<0.05, Kruskal-Wallis test followed by Nemenyi post hoc tests, n.s., not significant). See also *Figure 3—figure supplement 1* and *Figure 3—source data 1*.

The online version of this article includes the following source data and figure supplement(s) for figure 3:

**Source data 1.** The persistent state is specific to host seeking.

**Figure supplement 1.** Blood-fed mosquitoes respond to fictive sugar.

pulse of fictive $CO_2$ can influence the response to body heat and ultimately if it can induce blood feeding many minutes afterward.

First, we tested whether fictive $CO_2$ primes subsequent responses to heat (*Figure 4A*). Because $CO_2$ is highly volatile, mosquitoes likely sense this cue before body heat in naturalistic host-seeking settings. When we presented heat first followed by fictive $CO_2$, relatively small behavioral responses were evoked (*Figure 4B–C*). However, when fictive $CO_2$ was presented simultaneously with heat or 15 or 60 s prior to heat, larger walking and probing responses were seen (*Figure 4B–C*). This suggests that mosquitoes respond most strongly to the naturalistic temporal order of these host cues. Next, we asked how long prior to the heat stimulus fictive $CO_2$ can boost the response (*Figure 4D*). Compared to heat alone, walking and probing were detectably increased when a fictive $CO_2$ stimulus was presented up to 4 min prior (*Figure 4E–F*). This demonstrates that in addition to the sustained behavior response, fictive $CO_2$ increases the response to heat for minutes afterward.

Once a mosquito pierces the skin of a human host, taste cues present in the blood guide the decision to engorge. Because this is the final goal of host-seeking behavior, we wondered whether mosquitoes in the fictive $CO_2$-triggered persistent state had altered responses to both heat and taste stimuli. To test this, we designed a blood blanket assay that incorporated a thin sheet of a blood meal mimic located between the thermocycler heating element and the mesh below the mosquito, allowing it to be rapidly heated and cooled (*Figure 4G–H*). We used a solution of ATP in saline, which has previously been shown to be a highly palatable meal that induces females to engorge in the same manner as blood (*Galun et al., 1963*; *Jové et al., 2020*). This allowed us to test how the prolonged arousal state induced by fictive $CO_2$ influences the decision of the mosquito to feed on a blood meal mimic.

First, we explored the temporal relationship of host stimuli to test whether the naturalistic order of $CO_2$, heat, and taste stimuli elicits greater rates of feeding. When fictive $CO_2$ was presented along with heating of the meal for 10 min, many females fed to repletion (*Figure 4I*). To test the temporal order of these cues, we offered fictive $CO_2$ and a meal that was only warmed for several minutes, resulting in reduced feeding levels (*Figure 4I*). When the order was swapped and females were briefly offered the warm meal prior to fictive $CO_2$, very few females fed (*Figure 4I*). This suggests that mosquitoes feed at the highest rates when they receive a $CO_2$ stimulus prior to heat and taste stimuli.

Next, we asked how long after a brief pulse of fictive $CO_2$ females would retain the motivation to feed on the blood meal mimic. When offered the warm meal without fictive $CO_2$, few females fed. We then stimulated the females with fictive $CO_2$ and heated the blood meal mimic either immediately or after a delay of 2, 8, 14, or 20 min after fictive $CO_2$. We hypothesized that if the fictive $CO_2$ induced a persistent state of host seeking, it might trigger engorging behavior many minutes later. Indeed, fictive $CO_2$ was able to potentiate feeding when presented up to 14 min prior to heating of the blood meal mimic (*Figure 4J*). Taken together, our results demonstrate that the persistent host-seeking state increases host-seeking behaviors and alters the response to sensory cues for many minutes after a brief fictive $CO_2$ stimulus. We speculate that this reflects the amount of time a mosquito will pursue a host in a naturalistic setting before halting the search if it appears that the host is no longer nearby.

We noticed that there is considerable individual variation in how mosquitoes respond to the warm blood meal mimic after being activated by fictive $CO_2$ (*Figure 4J*). We asked whether the behavior state of individual mosquitoes could be inferred over longer periods of time. We extracted 38 behavior parameters from the experiment in *Figure 4J* from 30-s-long time windows and used t-distributed stochastic neighbor embedding (tSNE) to visualize the relationships. This embedding revealed that mosquito behavior fell into four major states that we termed rest, global search, local

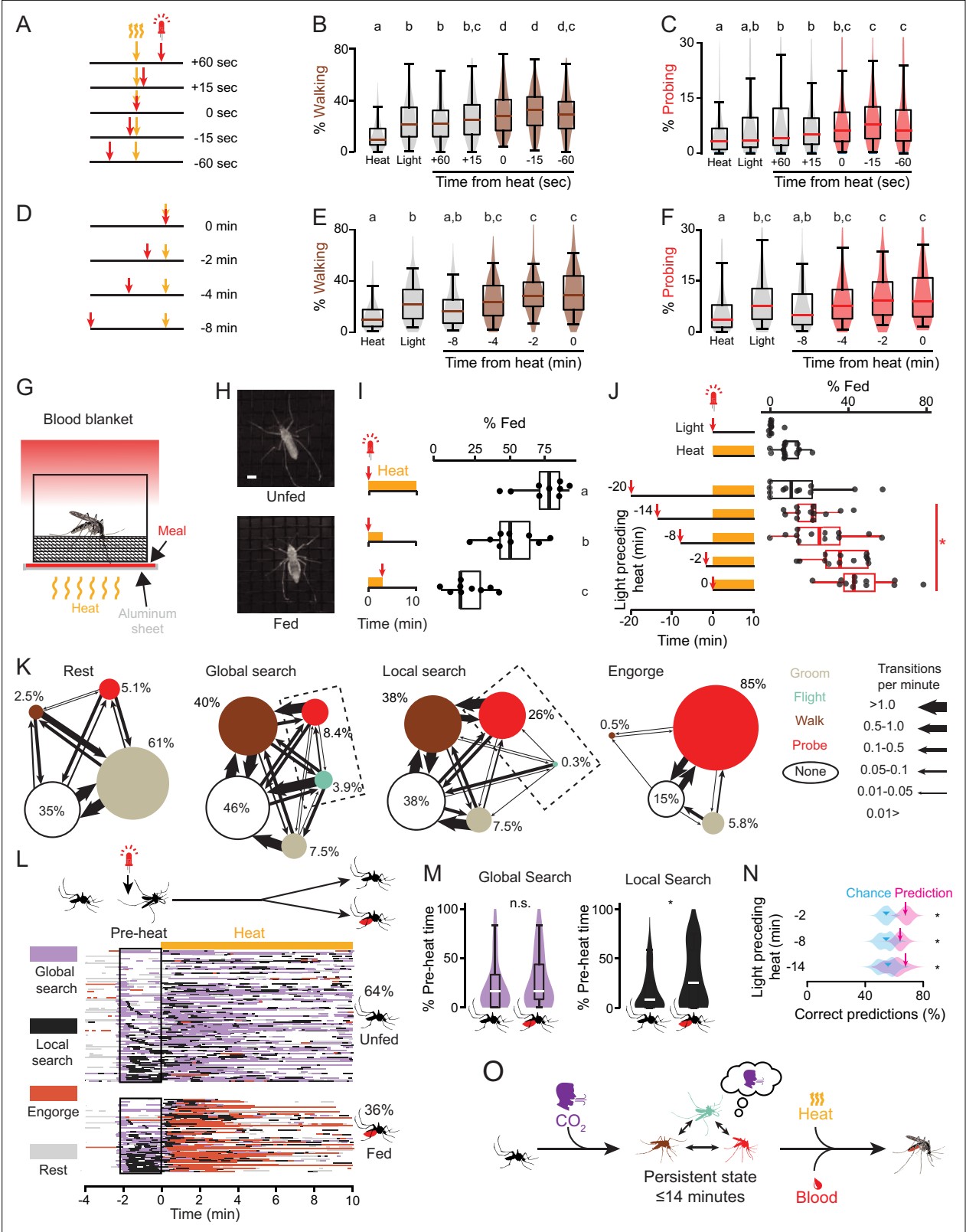

**Figure 4.** The persistent state integrates host cues and decision making in time. (**A–F**) Schematic of stimuli presentation (**A,D**) and quantification of walking (**B,E**) and probing (**C,E**) in the 5 min after the first stimulus onset (n=111–112 mosquitoes, 2 stimulus presentations/mosquito). In (**B,C,E,**and **F**), data are displayed as violin-box plots (median: horizontal line, interquartile range: box, 5th and 95th percentiles: whiskers). Data labelled with different letters are significantly different (p<0.05, Friedman test followed by Nemenyi post hoc tests). Data that are significantly different from heat

*Figure 4 continued on next page*

*Figure 4 continued*

or light are shaded in brown (**B**,**E**) or red (**C**,**F**). (**G**) Schematic of blood blanket assay which uses a blood meal mimic. (**H**) Still images of unfed (top) and fed (bottom) mosquitoes. Scale bar: 1 mm. (**I**) Percent mosquitoes engorged in response to the indicated stimuli in the blood blanket assay (n=9 trials/stimulus, 14 mosquitoes/trial). Data labelled with different letters are significantly different (p<0.05, ANOVA followed by Tukey's post hoc test). (**J**) Percent mosquitoes engorged in response to the indicated cues in the blood blanket assay (14 mosquitoes/trial n=11 trials/stimulus). Data in (**I**,**J**) are displayed as dot-box plots (median: horizontal line, interquartile range: box, 1.5 times interquartile range: whiskers). Dot-box plots in (**J**) with a red border signify data where combined light and heat stimuli are greater than the sum of individual stimuli (*p<0.05, Student's t-test, after adjustment for multiple comparisons using Holm's method). (**K**) Transition ethograms for each of the four states indicating the proportion of each behavior and the rate of transitions between them, with dashed rectangle highlighting difference between global and local search. (**L**) Inferred states of 168 individual mosquitoes from the 2 min pre-heat stimulus from experiment in (**J**), separated into those that were visually scored as unfed (top, n=108) or fed (bottom, n=60) at the end of the experiment. Each row represents data from one mosquito and data are sorted according to the amount of local search during the pre-heat period. White indicates none of the four states were inferred (i.e. the mosquito was primarily motionless). (**M**) Quantification of the percent of time mosquitoes spent in the indicated state during the 2 min pre-heat period. Data are plotted as violin-box plots (median: horizontal line, interquartile range: box, 5th and 95th percentiles: whiskers (*p<0.01), Mann-Whitney U-test, n.s., not significant). (**N**) Performance of a classifier trained on the proportion of each behavior in pre-heat period and used to predict feeding (magenta arrow) along with 10,000 bootstrapped classifiers (magenta violin plot). Chance value (cyan arrowhead) indicates the median performance of model on 10,000 shuffles (cyan violin plot) of the feeding data in (**J**). (n=166–168 mosquitoes/stimulus, p=2e-4, 0.0246, 0.0123 for 2, 8, and 14 min pre-heat periods, respectively, bootstrapped z-test.) (**O**) Summary of the persistent internal state for host-seeking behavior. Color of the mosquito silhouettes indicates the behavior depicted using the colors in **K**. See also *Figure 4—figure supplement 1*, *Figure 4—figure supplement 2*, and *Figure 4—source data 1*.

The online version of this article includes the following source data and figure supplement(s) for figure 4:

**Source data 1.** The persistent state integrates host cues and decision making in time.

**Figure supplement 1.** Inference of internal state from behavior.

**Figure supplement 2.** Mosquito states are consistent across window duration.

search, and engorge (*Figure 4K*, *Figure 4—figure supplement 1*). These states were observed even when the length of the time window was varied over a sixfold range (*Figure 4—figure supplement 2*). The states differed in the proportions of behaviors observed, the transition rates between them, and the total distance the mosquito travelled while in each state. We divided the data into those females that eventually fed or did not feed at the end of the experiment (*Figure 4L*). As expected, the engorge state was greatly enriched in mosquitoes that were categorized as fed, demonstrating that this method is effective at identifying longer-timescale behavior states (*Figure 4L*).

We asked whether the behavior state of the mosquito after fictive $CO_2$ was associated with the future decision to feed on the warm blood meal mimic. We found that fed and unfed mosquitoes showed similar amounts of global search pre-heat but greater levels of local search for those that would later feed (*Figure 4M*). The local search state differs primarily by showing more probing and less flight than the global search state. To avoid potential distortions created by the two-dimensional embedding (*Chari et al., 2021*), we trained a linear classifier using only the four behaviors (groom, walk, flight, and probe). A classifier trained on the proportion of time mosquitoes spent in each behavior for 2 min after the light stimulus could predict which mosquitoes would later feed with above-chance accuracy (*Figure 4N*). Thus, the behavior state that individual mosquitoes enter in response to fictive $CO_2$ reflects the likelihood of response to future sensory cues.

## Discussion

We investigated how female mosquitoes pursue humans by combining multisensory stimuli in time to achieve the goal of feeding on human blood. To precisely control the delivery of $CO_2$, we generated optogenetic tools to deliver fictive $CO_2$ and combined this with high-resolution behavior assays and machine learning analysis approaches. These experiments demonstrated that optogenetic activation of $CO_2$ sensory neurons induced a long-lasting behavior state change (*Figure 4O*). During this time mosquitoes had heightened responses to heat and were more likely to feed on a blood meal mimic even if encountered minutes after the $CO_2$ stimulus.

Integration of sensory information over time allows the nervous system to optimize decision making for a particular goal (*Körding, 2007*). Testing whether predators use this approach during hunting requires a precise understanding of what the predator is sensing and when, precluding field studies. Laboratory studies have found behavior responses altered for as long as a few seconds after encountering $CO_2$ in mosquitoes and other olfactory stimuli in *Drosophila* (*Alvarez-Salvado et al., 2018*;

*Dekker and Cardé, 2011*; *Demir et al., 2020*; *Pang et al., 2018*). Although this timescale of sensory integration is sufficient for upwind tracking, modelling suggests that integration over long timescales can maximize the information about the location of an olfactory stimulus (*Vergassola et al., 2007*). Here, we show that mosquitoes integrate olfactory, heat, and taste stimuli for at least 14 min, much longer than previously assumed. The increased movement and the bias of actions toward a particular goal constitute an internal motivational state sustained over minutes—one specific to feeding on humans. Importantly, our results show that this internal state does not require constant flight behavior or a constant air flow stimulus, demonstrating that it is maintained by an internal mechanism rather than continuous sensory or reafferent stimuli.

In principle, the long-duration characteristic of the host-seeking state could be generated by neurons at any point in the circuit. However, it is extensively documented in *A. aegypti* that $CO_2$ sensory neurons show an accurate readout of $CO_2$ levels over a large concentration range (*Grant and O'Connell, 2007*; *Grant et al., 1995*). These neurons do not show adaptation or prolonged activity (*Grant et al., 1995*) and have similar responses in males and females (*Grant and O'Connell, 2007*). While we suspect, based on precedents for persistent states observed in other systems, that the persistent state is controlled in the central brain, we cannot exclude a contribution from the periphery. In mice, circuits originating in the central amygdala promote pursuit and attack during cricket hunting (*Han et al., 2017*), but these behaviors appear time-locked to optogenetic activation and a circuit controlling hunting persistence has not been identified.

Mosquito host seeking shares some characteristics with other social and feeding states (*Asahina et al., 2014*; *Flavell et al., 2013*; *Kohatsu et al., 2011*; *Hindmarsh Sten et al., 2021*; *Marques et al., 2020*). The pursuit of females by males during courtship behavior in *Drosophila* shows especially striking similarities. In response to female sensory cues or stimulation of a subset of *fruitless* neurons, male flies enter a state of increased courtship behaviors and lower thresholds for sensory cues from females (*Kohatsu et al., 2011*; *Clowney et al., 2015*; *Hindmarsh Sten et al., 2021*; *Inagaki et al., 2014*; *Jung et al., 2020*). It appears that this function of *fruitless* is conserved and displays the properties of persistence and sexual dimorphism (*Bertossa et al., 2009*; *Demir and Dickson, 2005*; *Manoli et al., 2005*). These similarities suggest that mosquito evolution may have co-opted these properties of ancestral *fruitless* circuits to drive a novel feeding behavior.

This study illuminates why mosquitoes are such effective predators: they maintain the goal of blood feeding for minutes even in the absence of any additional positive stimuli or reinforcement. Because this state greatly outlasts individual sensory stimuli and integrates multiple modalities, any intervention that disrupts this internal drive state should be more effective than vector control measures that mask or disrupt any individual aspect of host seeking.

# Materials and methods

### Key resources table

| Reagent type (species) or resource | Designation | Source or reference | Identifiers | Additional information |
|---|---|---|---|---|
| Recombinant DNA reagent | QUAS-CsChrimson-tdTomato (plasmid) | This study | Addgene RRID:175548 | Described in Materials and methods section 'Creation of CsChrimson mosquitoes for optogenetics'; available from Addgene |
| Strain, strain background (*Aedes aegypti*) | Liverpool IB12 | BEI resources | MRA-735 | |
| Genetic reagent (*Aedes aegypti*) | QUAS-CsChrimson-tdTomato | This study | | Strain available on request from Vosshall or Sorrells labs |
| Genetic reagent (*Aedes aegypti*) | Gr3-QF2 | doi:https://doi.org/10.1101/2020.11.07.368720 | | |
| Genetic reagent (*Aedes aegypti*) | Gr4-QF2 | PMID:33049200 | | |
| Antibody | Mouse monoclonal anti-Brp antibody | DSHB | Cat# nc82, RRID:AB_23 14866 | Immunofluorescence 1:10,000 |
| Antibody | Rabbit polyclonal anti-RFP antibody | Rockland | Cat# 600401-379, RRID:AB_22 09751 | Immunofluorescence 1:1000 |

*Continued on next page*

*Continued*

| Reagent type (species) or resource | Designation | Source or reference | Identifiers | Additional information |
|---|---|---|---|---|
| Antibody | Goat polyclonal anti-mouse Alexa Fluor 647 | Thermo Fisher | Cat# A21235, RRID:AB_25 35804 | Immunofluorescence 1:500 |
| Antibody | Goat polyclonal anti-rabbit Alexa Fluor 555 | Thermo Fisher | Cat# A32732, RRID:AB_26 33281 | Immunofluorescence 1:500 |
| Software, algorithm | Ctrax | PMID:19412169 | | |
| Software, algorithm | JAABA | PMID:23202433 | | Downloaded July 15, 2020 |
| Software, algorithm | Animal Part Tracker (APT) | https://github.com/kristinbranson/APT | | Downloaded July 9, 2020 |

## Mosquito strains

The following *A. aegypti* strains were used in this paper: wild-type Liverpool, *Gr3-QF2* (*Younger et al., 2022*), *Gr4-QF2* (*Jové et al., 2020*), fruitless$^{\Delta M}$ (*Basrur et al., 2020*), fruitless$^{\Delta M\text{-tdTomato}}$ (*Basrur et al., 2020*), and *QUAS-CsChrimson* (this study).

## Mosquito rearing

Mosquito strains were reared at 26°C ± 2°C with 80% humidity and 14 hr light, 10 hr dark (lights on at 7 AM) as previously described (*DeGennaro et al., 2013*). Embryos were hatched in hatching broth: 1 pellet of fish food (TetraMin Tropical Tablets, Pet Mountain 16110M) crushed using a mortar and pestle, added to 850 mL deionized water, then autoclaved. Larvae were reared in deionized water and fed 1–3 tablets of fish food per day. Adult mosquitoes were fed on 10% sucrose (w/v in distilled water) ad libitum. Sucrose was delivered in a Boston clear round 60 mL glass bottle (Fisher FB02911944) filled with 50 mL 10% sucrose. A cotton dental wick (Richmond Dental 201205) was inserted into the bottle and mosquitoes fed from the sugar-moistened wick. Female mosquitoes were blood fed on mice or human arm to generate eggs. Eggs were dried at 26°C and 80% humidity for 3 days, and then stored at ambient temperature and humidity for up to 3 months. Adults were allowed to mate freely for at least 7 days prior to performing experiments. All behavior experiments were carried out in the light phase of the photoperiod, with most experiments occurring between zeitgeber (ZT) ZT2 and ZT12.

## Creation of CsChrimson mosquitoes for optogenetics

We generated mosquitoes that expressed a translational fusion of CsChrimson to tdTomato under the control of the QUAS promoter, referred to as *CsChrimson* or *QUAS-CsChrimson* throughout the paper. The coding sequence of *CsChrimson-tdTomato* was PCR-amplified from the vector p20X (*Klapoetke et al., 2014*) using the following oligonucleotide primers: forward 5'- CTCGAGCAAAAT GAGCAGACTGGTCGCCGCTTC-3', reverse 5'- ATCCTCTAGATTACACCTCGTTCTCGTAGCAGAATT TATACAG-3'. The vector backbone from pXL-BacII-15xQUAS_TATA-SV40 (*Riabinina et al., 2016*) was amplified by PCR using the following oligonucleotide primers: forward 5'-GTCTGCTCATTTTGCT CGAGCCGCGGCCGCAGATC-3', reverse 5'-CGAGGTGTAATCTAGAGGATCTTTGTGAAGGAACCT TACTTCTG-3'. The *CsChrimson* insert was cloned into the backbone using Infusion HD cloning kit (Takara 638920) to create pTS26, available at Addgene (plasmid number 175548). This plasmid was injected into 500 *A. aegypti* Liverpool embryos by the Insect Transformation Facility (Rockville, MD) using 200 ng/µL plasmid DNA and 200 ng/µL piggyBac transposase mRNA. Ten independent *QUAS-CsChrimson* integration events were isolated under standard mosquito rearing conditions.

## Peripheral sensory appendage microscopy

Mosquitoes 3–4 weeks of age were anesthetized on ice, then maxillary palps and antennae were removed using sharp forceps and placed in fixative (4% paraformaldehyde, 0.1 M Millonig's Phosphate Buffer pH 7.4, 0.25% Triton X-100) and nutated for 30 min at 4°C. Tissues were washed four times in PBS, then mounted in SlowFade Diamond Antifade Mountant (Thermo Fisher). Images were acquired on an Inverted LSM 780 laser scanning confocal microscope (Zeiss) using a 25 × 0.8 NA multi-immersion objective with oil. Images were processed using ImageJ.

## Brain immunostaining

Brain immunostaining was carried out as previously described (*Jové et al., 2020*). Mosquitoes 1–2 weeks of age were anesthetized on ice, then heads were removed using forceps and placed into

fixative (4% paraformaldehyde, 0.1 M Millonig's Phosphate Buffer pH 7.4, 0.25% Triton X-100) and nutated for 3 hr at 4°C. Heads were washed four times in PBS and kept on ice during dissections. The brains were dissected using #5 forceps (Dumont) in a droplet of PBS on a Petri dish coated with SYLGARD silicone elastomer (Dow). Brains were transferred to a 35 µm mesh cap of a flow cytometry test tube (Fisher 08-771-23) in a 24-well plate containing PBSTx (PBS with 0.25% Triton X-100). Brains were washed four times for 30–60 min at room temperature in PBSTx on an orbital shaker before permeabilization and between each of the following steps. Brains were permeabilized in PBS with 4% Triton X-100% and 2% normal goat serum for 2 days at 4°C on an orbital shaker. We used the mouse anti-Bruchpilot (brp) monoclonal antibody at a dilution of 1:10,000. The brp antibody was purified by Frances Weis-Garcia of the Sloan Kettering Institute Antibody & Bioresource from the brp/nc82 hybridoma, developed by Erich Buchner at the Universitätsklinikum Würzburg and obtained from the Developmental Studies Hybridoma Bank, created by the NICHD of the NIH and maintained at The University of Iowa, Department of Biology, Iowa City, IA. Rabbit anti-RFP antibody (Rockland 600-401-379) was used at a dilution of 1:1000 to detect tdTomato fused to CsChrimson as well as the dsRed transgene marker expressed from the *3XP3* enhancer/promoter. Brains were incubated in primary antibodies in PBSTx with 2% normal goat serum for 3 days at 4°C on an orbital shaker. Secondary antibodies were goat anti-mouse Alexa Fluor 647 (Thermo Fisher A21235) and goat anti-rabbit Alexa Fluor 555 (Thermo Fisher A32732) both at 1:500 dilution. Brains were incubated in secondary antibodies in PBSTx and 2% normal goat serum for 2 days at 4°C on an orbital shaker. Brains were washed four more times at room temperature for 30–60 min before mounting in SlowFade Diamond Antifade Mountant (Thermo Fisher). Images were acquired on an Inverted LSM 880 Airyscan NLO laser scanning confocal microscope (Zeiss) using a 25 × 0.8 NA multi-immersion objective with oil. Images were processed using ImageJ.

## Rearing mosquitoes for optogenetics

For CsChrimson to respond to red light (625 nm in this paper), it is necessary to supply the all-trans retinal co-factor. Moreover, it is critical that animals reared for optogenetics be maintained in the dark to avoid activating CsChrimson inappropriately. Therefore, we developed a mosquito rearing protocol to deliver all-trans retinal under dark conditions. First, we carried out experiments to select the best *QUAS-CsChrimson* transgenic insertion among the 10 lines we generated. Insertions were identified and tracked using fluorescence from the *3xP3-ECFP* marker. We wanted lines with strong and selective behavioral induction in combination with a QF2 driver, but no basal behavioral activity without a QF2 driver. We also wanted it to be a single insertion at a known position in the genome, that would not obviously disrupt a known gene. The insertion site of the transgene in each line was mapped to the genome using TagMapping (*Stern, 2016*). After being fed with all-trans retinal as described below, all 10 lines were tested for their response to red light with and without being crossed to a QF2 driver. Based on these initial screens, a single line with the *QUAS-CsChrimson* transgene inserted in an intron of the gene LOC23687794 on chromosome 2 at base pair 453,953,698 in the L5.0 version of the *A. aegypti* genome (*Matthews et al., 2018*) was selected for use in all subsequent experiments. This *CsChrimson* strain was outcrossed to wild-type mosquitoes for eight generations before being homozygosed and used for behavior experiments. For all experiments except those in *Figure 2I and K*, which used wild-type mosquitoes with a green light startle stimulus, animals were subjected to special rearing conditions to prepare them for optogenetics. Eggs were hatched in 1 L of hatching broth under a 14 hr 450 nm blue light and a 10 hr dark cycle in a light-tight 26°C incubator with 80% humidity. Blue light was selected for the light phase of the photoperiod to avoid activating CsChrimson. The next day 2 L of distilled water was added to the pan, and the following day larvae were thinned to 450 per pan. Larvae were subsequently sorted for fluorescence markers if necessary, using a dissecting microscope. Larvae were fed daily with 1–3 tablets of Tetramin fish food (Pet Mountain 16110M) ground into a powder using a mortar and pestle. Pupae were moved to eclose into adults in a 30 cm × 30 cm × 30 cm insect rearing cage (Bugdorm DP1000) with ad libitum access to 10% sucrose in sugar-feeding glass bottles. Animals were not sexed at this stage, so cages contained males and females that freely mated. Behavior experiments were performed 1–4 weeks post-pupation. Three days before the experiment, the sugar wicks were replaced with water wicks to starve animals for 24 hr. Two days before the experiment, the water wicks were replaced with wicks soaked in 10% sucrose and 400 µM all-trans retinal (Sigma, R2500-1G). Fifty mL of sucrose and

all-trans retinal was used per cage. Animals were allowed to feed for 1–3 days in the dark on this meal. In pilot experiments we verified that starved females fed on sucrose and all-trans retinal by observing yellow pigmentation in the abdomen. Feeding in the dark was used to avoid premature neuronal activation and bleaching of the all-trans retinal in the sugar feeders. This rearing protocol was used for all experiments in the opto-membrane feeder, opto-thermocycler, and blood blanket experiments.

## Opto-membrane feeder assay

The opto-membrane feeder assay was constructed using optomechanical components (Thorlabs MB12, TR12, RA90) and a black 1/4" thick acrylic platform for the canister of mosquitoes to rest on. A hole in the bottom of the platform allowed a camera (Blackfly U3-13S2M-CS, FLIR) outfitted with an 800 nm longpass filter (Midwest Optical LP800-34) to image through the clear canister. Canisters were constructed from a polycarbonate tube of diameter 4.5" (McMaster-Carr 8585K56) and 5" long. The bottom was made of clear 1/8" thick acrylic and attached with plastic epoxy (Loctite 1363118). The top was an inset lid made of black 1/4" and 1/8" acrylic and UV-resistant black mesh (McMaster-Carr 87655K13). The canister was surrounded by a coil of RGB LEDs (Digikey 289-1189-ND) spaced 1.5" from the exterior of the canister and controlled by an Arduino Uno board (Arduino A000066). Mosquitoes were illuminated by 850 nm infrared LEDs surrounding the top of the cylinder of RGB LEDs. The assay was enclosed in a black 1/4" thick acrylic box of dimensions 15" × 15" × 28" to prevent ambient light from entering the assay. The top of the acrylic box had an entry port of 4" × 2.7" for $CO_2$ diffused by a Flystuff Flypad (Genesee Scientific, 59-114), and two doors on the side, one at the level of the cylinder (10" high × 8" wide) and one at the bottom (8" high × 10" wide) at the level of the camera. The day before the experiment, mosquitoes were sexed under cold anesthesia in white light, placed into the cylindrical canisters, and fed water and 400 μM all-trans retinal in the dark until the experiment commenced. Dental wicks were soaked in approximately 12.5 mL of the water and all-trans retinal, placed on top of the mesh of the inset lid. Trials were run in an environmental room at 25–28°C and 70–80% humidity. For each trial, a canister of 20 mosquitoes was placed on the platform and acclimated for 10 min prior to the stimulus. Throughout the acclimation period and trial, the canister was bathed in dim blue 471 nm light from the RGB LEDs. The RGB LEDs were arranged in a coil around the cylindrical canister of mosquitoes to give a relatively even light distribution throughout the canister as measured by a light meter (Coherent Wand UV/VIS Power Sensor 1299161). At the start of the trial, a blood meal consisting of 5 mL of defibrinated sheep blood (Hemostat Laboratories DSB100) with 2 mM ATP (Sigma A6419-1G) heated to 42°C was placed on top of the canister. Blood meals were delivered using an acrylic lid consisting of a 1/16" thick clear acrylic ring with a 2" inner diameter and 2.6" outer diameter attached to a 1/2" thick clear acrylic ring with a 2.3" inner diameter and 2.6" outer diameter. This lid was covered with Parafilm on the 1/16" thick side to create a well for the blood when placed Parafilm-down on the inset lid of the cylindrical canister. At the start of the experiment, the warm blood meal was pipetted onto the Parafilm. On top of the blood meal was an inverted 4 oz bottle (SKS Bottle & Packaging 0604–07) filled with water heated to 42°C to keep the blood near body temperature for the duration of the 15 min trial. Mosquitoes were given $CO_2$, red light (624 nm, 3.5–6 μW/mm$^2$), or neither stimulus throughout the 15 min trial. The light intensity chosen was an intermediate intensity as determined by a light-behavior dose-response curve (*Figure 1—figure supplement 1B-D*); 10% $CO_2$ was mixed with filtered room air using flow controllers (Aalborg P26A1-BA2) to deliver a 2.7% $CO_2$ stimulus through the top of the container. Air flow delivery is described in detail in *Basrur et al., 2020*. Between trials, the lower door was opened for 5 min with the air flow on to flush residual $CO_2$ from the assay. On a given day of experiments, each of three genotypes (*Gr3*, *CsChrimson*, and *Gr3 > CsChrimson*) was tested with each of the three stimuli (no stimulus, $CO_2$, red light), for a total of nine trials. The order of trials was rotated between days. Genotypes were blinded to the experimenter. Attraction to the warm blood meal was quantified by manually counting the number of mosquitoes on the warm blood meal in the video (1 frame/second) every 15 s. Engorgement was quantified by visual examination of mosquitoes at 4°C after the end of the trial. Between days of experiments, the canisters were cleaned by spraying 70% ethanol with a spray bottle and wiping down with a soft sponge, rinsed with deionized water, and air dried.

## Opto-thermocycler assay

The opto-thermocycler assay was constructed on top of a PCR thermocycler (Eppendorf Mastercycler) using optomechanical components (Thorlabs XE25L12, XE25L24, XE25L09, XE25T4, RA90, TRA6, TR12). This assay was used as the basis for experiments delivering light only, light along with heat stimuli, and blood blanket experiments. Light was delivered using six red light 627 nm LEDs (Luxeon Star SP-01-D9) or six green light 530 nm LEDs (Luxeon Star SP-01-G4) controlled with an Arduino Uno board. Light intensity was measured using a Coherent Wand UV/VIS Power Sensor (1299161) at nine points on the surface of the PCR block and light angle was adjusted using wires to achieve even illumination. The surface of the PCR block was covered in black tape to reduce glare (Thorlabs T137-2.0). Temperature was measured using a type T thermocouple (Harold G Schaevitz Industries LLC CPTC-120-X-N) connected to the Arduino board (Arduino A000066) using a thermocouple amplifier (Adafruit MAX31856). The thermocouple sensor was placed on the surface of the lower right of the PCR block and secured using black tape (Thorlabs T137-2.0). Temperature reading and light output were recorded every 100 ms from the Arduino using a custom Processing script. Video was synchronized with the light and temperature stimuli with an infrared 940 nm LED (Adafruit 387) covered with tape and placed in the field of view of the camera. Mosquitoes were illuminated with an infrared 850 nm LED strip (Waveform Lighting 7031.85) surrounding the plate of mosquitoes orthogonal to the view of the camera. Video was recorded using a Blackfly camera (FLIR BFS-U3-16S2M-CS) outfitted with a 780 nm longpass filter (Vision Light Tech LP780-25.5) at 30 frames/s using Spinview software. Heat stimuli were programmed onto the PCR thermocycler to elicit the desired change in temperature from ambient to skin temperature (25–35°C) as measured by the thermocouple (*Figure 2B*). Red light stimuli were 627 nm at an intensity of 12 µW/mm$^2$, chosen as an intermediate intensity that allowed the possibility of both an increase and a decrease in the behavioral response. Green light stimuli were 530 nm at an intensity of 22 µW/mm$^2$, chosen because it was the maximum intensity of our setup and to maximize the chance of observing a persistent response to green light. To synchronize the heat and light stimuli, experiments started with a brief dip in temperature followed by a 10 min acclimation period after which the experiment started. Experiments in *Figure 2E–H* were conducted with a single stimulus presented to mosquitoes to determine the duration of response. In all other experiments, mosquitoes received multiple stimuli over the course of a 3–6 hr experiment. Data from the rare trials where the mosquito died during the experiment were discarded. For experiments using red light and heat, trials were delivered 20 min apart and the order was pseudorandomized between multiple sweeps of trials and across days. For experiments using green light and heat (*Figure 2I and K*), the stimuli were pseudorandomized across sweeps only. Sweeps are considered technical replicates conducted on the same individual mosquito, and each mosquito in the assay is a biological replicate. All experiments were conducted multiple times either as a pilot followed by full experiment or multiple full experiments. Mosquitoes were assayed in a custom acrylic plate with 3 × 5 wells. The sides of the plate were cut using a laser cutter from 1/8" thick clear acrylic, then assembled using acrylic glue (WELD-ON, #4SC Plastic Solvent Glue for Acrylic). The top and bottom were cut from 1/16" acrylic. The top was left removable to load mosquitoes while the bottom was used to sandwich a piece of black fiberglass window screen (Breakthrough Premium Products IHLRS3684BL) creating a mesh bottom for each well. The acrylic bottom piece spaced the mesh bottom of the wells 1.5 mm from the surface of the PCR block. Wells containing the mosquitoes were 18.5 mm long × 17 mm wide × 12 mm high. The well in the lower right was empty to accommodate the thermocouple. The day before the experiment, mosquitoes were sexed under cold anesthesia in white light, placed into the custom plate, and fed water and 400 µM all-trans retinal overnight until the experiment. This was delivered in cotton dental wicks each soaked with 12.5 mL water and all-trans retinal. Three wicks were laid flat beneath each plate so that mosquitoes in all wells could access the wicks beneath. Experiments were run at ambient room temperature and humidity, but the PCR block kept the assay chamber at a fixed temperature. Between trials the surface of the PCR block was cleaned by wiping with a Kimwipe moistened with 70% ethanol. Between days of experiments, the acrylic plates were cleaned by spraying 70% ethanol with a spray bottle and wiping down with a gloved finger, rinsed with deionized water, and air dried.

## Blood blanket assay

The opto-thermocycler assay captures probing behavior but does not offer a meal for engorgement. We therefore modified this device to produce the blood blanket assay. The most biologically relevant meal for host-seeking females would be blood, but its opacity makes it unsuitable for our video tracking. We therefore used adenosine 5'-triphosphate (ATP) in saline as a an optically clear proxy for blood. This meal has previously been shown to be highly palatable and triggers mosquito engorgement equivalent to a blood meal (*Galun et al., 1963*; *Jové et al., 2020*). To modify the opto-thermocycler to accommodate this blood meal substitute, a thin aluminum plate (McMaster-Carr 6061 Aluminum sheet 0.025") was sandwiched between laser cut pieces of acrylic creating wells on the side facing the mosquito. The wells were 18.5 mm × 17 mm × 1 mm. The acrylic was bonded to itself using acrylic glue (WELD-ON, #4SC Plastic Solvent Glue for Acrylic) and to the aluminum plate with epoxy (Loctite 1363118) and UV-curing glue (Bondic SK8024). The plate was prepared for a trial by adding 500 µL of the meal (110 mM NaCl, 20 mM $NaHCO_3$, and 2 mM ATP) to each well of the plate in *Figure 4A–C*. In *Figure 4D–F*, the composition of the meal was 110 mM NaCl, 10 mM $NaHCO_3$, and 2 mM ATP. The plate was covered with Parafilm to provide a membrane for the mosquitoes to pierce before accessing the meal. The plate was placed directly on top of the PCR block to allow maximum heat transfer. The thermocouple was placed on the surface of the Parafilm in the middle of the well in the lower right corner to record the temperature of the heated meal. Trials were carried out and synchronized in the same way as opto-thermocycler experiments. All blood blanket experiments were single trial.

## Machine learning-based behavior classification

Videos were pre-processed using a custom Python script tracking_optothermo.py that converted the file format, split up videos into ~30 min chunks, selected frames to create a background image for centroid tracking, and detected frames where the IR synchronization LED was illuminated. Next, we used Ctrax (*Branson et al., 2009*) for centroid tracking. A background model was created using the selected frames from the experimental video. Ctrax background settings were background brightness high threshold 2.55, low threshold 0.25–0.5 adjusted depending on the video. The area with the infrared synchronization LED was excluded using a region of interest to avoid interference with the tracking. Mosquitoes that moved very little, such that they were visible in the background image, were corrected for using the Fix Background Model option. In tracking settings, shapes were filtered using the following minimum/maximum: 110/1600 for area, 4/36 for major axis, 4/30 for minor axis, 0.0/0.98 for eccentricity. The rest of the tracking settings were default. We used Ctrax centroid tracking as input to APT (https://github.com/kristinbranson/APT downloaded on July 9, 2020; *Branson, 2022*) for tracking points on the mosquito body. For opto-thermocycler experiments, we tracked nine points: the tip and base of the proboscis, the tip of the abdomen, and three points on each foreleg: where the femur connects to the body, the joint between the femur and the tibia, and the joint between the tibia and the first tarsomere. Tracking the mid legs and hind legs was not needed for behavioral classification so they were excluded to speed computation. Opto-thermocycler classifiers were trained on 320 frames from two videos for female mosquitoes and 102 frames from one video for male mosquitoes. We tracked 13 points for blood blanket experiments, the same nine points as for opto-thermocycler experiments plus two points at the point of the abdomen where it connects to the thorax and two points at the midpoint or thickest part of the abdomen. The blood blanket classifier was trained on 215 frames from four videos. All APT classifiers were trained using the Cascaded Pose Regression tracking algorithm. JAABA (*Kabra et al., 2013*) (downloaded on July 15, 2020) was used for classifying specific behaviors. The classifier for flight (called *fly2*) was used for all videos of females and males. It was trained from two videos and used appearance and locomotion features with radius of 10 frames with no post-processing. The other classifiers additionally used APT information, a larger radius of frames, and minimum bout sizes for improved accuracy. Separate classifiers were trained for females and males in the opto-thermocycler and females in the blood blanket experiments to maximize classifier accuracy in the face of differences in visual appearance. Probing classifiers (*probe5* for female opto-thermocycler experiments, *probemale* for male opto-thermocycler experiments, and *probeBB* for females in the blood blanket experiments) included the pair of points proboscis tip and base as features, along with APT, motion, and appearance features. The grooming and walking classifiers (*walk3* for female and male opto-thermocycler experiments, *groom3* and *groommale* for

female and male opto-thermocycler experiments, respectively, and *walkBB* and *groomBB* for blood blanket experiments) were trained using APT, locomotor, and appearance features. APT classifiers were visually inspected for accuracy. APT and JAABA classifiers were evaluated by the accuracy of ground truthing on the JAABA classifiers. An initial classifier was trained, then ground truthing was performed on 50–100 segments of 1 s video segments that were balanced between segments with and without the behavior. These segments were examined for mis-classified frames and additional training was performed to improve the classifier. Thus, the ground truth dataset is more challenging than a random one because it contains frames that were previously mis-classified and so the real accuracy is higher. Training continued until true positive and true negative rates of >90% were obtained with seven of nine classifiers. Two other classifiers had rates slightly below this. The *groomBB* was trained to ~84% true positive and negative rates because certain grooming postures are difficult to distinguish from probing postures. The *probemale* classifier was trained to ~87% true positive and ~91% true negative rates because only part of the male proboscis is distinguishable from the maxillary palps during probing behavior. Classified behaviors for each mosquito track from JAABA were assigned to single wells according to x–y location of the track to correct the small numbers of frames where Ctrax detected two mosquitoes per well (usually due to a leg that was discontinuous with the rest of the animal) and to connect broken tracks to a single individual mosquito. The IR LED stimulus in the video was aligned with data about temperature and light stimuli from the Arduino and assigned to frames in the video. Velocity was calculated by taking the Ctrax x–y position at 100 ms intervals (three frames).

## Analysis of behavior

To calculate the half-life of the mosquito behavior response in *Figures 2H–J, 3C and H*, the baseline was calculated as the average probing in 2 min prior to stimulus onset. A sliding window of the amount of probing was calculated in 15 s windows starting at stimulus onset for every frame. The maximum response was defined as the window with the greatest probing after stimulus onset and $t_{1/2}$ was defined as the first window in which the probing was halfway between the maximum response and the baseline. To calculate the integration of heat and the second stimuli (fictive $CO_2$, fictive sugar, or green light) in *Figure 2—figure supplement 2* and *Figure 2M*, we calculated the average response to each of the individual stimuli. We added the two responses to get a predicted additive response. For each individual mosquito, we divided its response by the predicted additive response and multiplied by 100%. This gave a percent additivity where 0% was no response and 100% was exactly additive. For line graphs, the additivity signal was smoothed over 4.5 s around each 500 ms timepoint.

## tSNE analysis

To infer the state of individual mosquitoes in the blood blanket experiment, we split each mosquito track into 30 s intervals at 10 s step size and calculated 38 parameters. The 30 s time interval was selected as a period of time over which the behaviors exhibited were relevant to interpreting the internal state of the mosquito. The time interval was varied from 10 to 60 s to assure that the results were not sensitive to this parameter choice (*Figure 4—figure supplement 2*). The parameters included the proportion of the time window that mosquitoes exhibited each behavior and no behavior. Mosquitoes can probe and walk at the same time so the proportion of time probing and walking, probing not walking, and walking not probing were included. The number of bouts of each behavior was included. Velocity parameters included average velocity over the window and average velocity during each behavior. Transitions between behaviors were included as outgoing rate per second of transition to all other behaviors or no behavior. For the purposes of transitions and ethograms, probing and walking were treated as mutually exclusive with probing taking higher precedence over walking. For all behaviors to avoid rare frames where multiple behaviors were classified for a single frame, the precedence of behaviors were flying > probing > walking > grooming > no behavior. Based on the total amount of time animals spent performing each behavior, cutoffs were determined to specify a minimum amount of behavior exhibited. Cutoffs were 0.04 for flight, 0.2 for walking or probing, and 0.3 for grooming. Behavior below these cutoffs was excluded from further analysis. The Python package scikit-learn (https://scikit-learn.org/ version 0.24.1) was used for tSNE with parameters n/100 perplexity (1061), and other default parameters (200 learning rate, 1000 iterations). Multiple perplexities were compared to assure that results were not sensitive to this parameter choice. tSNE plots

were examined and clusters were segmented manually by grouping densely clustered points. These clusters were used to annotate videos for visual inspection of what mosquito behaviors they corresponded to. Names for clusters were chosen based on the characteristics of the clusters shown in *Figure 4K*, *Figure 4—figure supplement 1B*, *Figure 4—figure supplement 2*, and video observation. Clusters that included mosquitoes that moved around were named Global or Local search based on the total amount of movement and contrasting amounts of flight and walk behaviors. The cluster that included mostly grooming was termed Rest. The cluster that included mosquitoes that were stationary, probing, and with abdomens expanded from feeding was termed Engorge. The clusters for Rest, Global Search, and Local Search were single clusters that were clearly differentiated on the tSNE. The Engorge cluster was composed of two smaller clusters that, when observed on video, both consisted of mosquitoes engorging and were therefore combined. Points on the end of the Local Search cluster in the tSNE with high probing were also examined by video and grouping was kept with the Local Search cluster.

## Statistical analyses

R (https://www.r-project.org version 4.0.5) and Python were used for statistical analysis. Data distributions were visually examined for normality or tested using the Shapiro-Wilk test. Normally distributed samples were compared by one sample t-test for paired measurements or ANOVA and Tukey's test for multiple categories. Non-normally distributed samples were compared using the Friedman test for multiple category repeated measurements, Kruskal-Wallis test for multiple category single measurements, or the sign test for skewed paired measurements. The Friedman and Kruskal-Wallis test were used with Nemenyi post hoc tests to determine pairwise differences between categories. t-Tests and sign tests were adjusted using Holm's method for correcting for multiple comparisons. For statistical analyses involving comparisons of the behavior of males and females, we repeated the tests after accounting for differences in classifier accuracy by changing the proportion of behavior by this difference (i.e. 4.93% for probing and 3.25% for walking) and confirming that the results were the same. Sample sizes followed conventions in the field. For experiments with multiple stimuli presented to each animal, 4–6 days of data were collected. For endpoint and single stimulus experiments, 7–11 days of data were collected.

## Logistic regression

Logistic regression models for the blood blanket experiment were trained using the Python sklearn package with the proportion of time mosquitoes spent in each of the four behaviors (groom, walk, probe, and fly) for 2 min after the light stimulus as predictors. These periods of time were –2 to 0, –8 to –6, and –14 to –12 min relative to the heat stimulus for the 2, 8, and 14 min inter-stimulus interval experiments. The dependent variable was whether the mosquito engorged by the end of the experiment. Models used the liblinear solver, random_state of 0, and balanced weight_class. Bootstrapping was performed using 10,000 resamples with replacement of the engorgement dataset to determine the distribution of predictive models; 10,000 shuffles of the engorgement data were used to determine whether the predictive model performed above chance. Leave-one-out cross-validation was used to determine whether the model was overfitted.

## Data availability

All data generated or analyzed during this study are included in the manuscript and in *Figure 1— source data 1*, *Figure 2—source data 1*, *Figure 3—source data 1*, and *Figure 4—source data 1*. Large datasets are available at https://github.com/trevorsorrells/Optothermocycler, (copy archived at swh:1:rev:9c70d0c348ad6d8a32663ca23257616f915ab06e; *Sorrells, 2021*).

## Code availability

Analysis code used in this publication is available at https://github.com/trevorsorrells/Optothermocycler; *Sorrells, 2021*.

## Acknowledgements

We thank Hessam Akhlaghpour, Josie Clowney, Emily Dennis, Ann Kennedy, Philip Kidd, and members of the Vosshall lab for comments on the manuscript. We thank Allen Lee, Alice Robie, and Kristin

Branson for sharing their unpublished APT and providing technical help with running it. We thank Jason Banfelder and Rebecca Bennett with software setup on the Rockefeller High Performance Computing Cluster; Jim Petrillo, Dan Gross, and Peer Strogies at the Rockefeller Precision Instrumentation Technologies facility for assistance with design and fabrication of behavior assays; Gloria Gordon and Libby Mejia for expert mosquito rearing; Tom Hindmarsh Sten, Veronica Jové, Gaby Maimon, Ben Matthews Chris Potter, Vanessa Ruta, and Nilay Yapici for discussions; Annie Handler, Jazz Weisman, and Ari Zolin for technical advice with optogenetics experiments; Ben Matthews, Meg Younger, and the Aedes Toolkit Group for access to unpublished strains; Cong Li for assisting with an early version of the individual state analysis; and Rob Harrell at the Insect Transformation Facility for embryo injections. Pilot experiments for this study used an optogenetic setup in the lab of Vanessa Ruta. This work was funded by a Jane Coffin Childs postdoctoral Fellowship (TRS), and a Kavli Neural Systems Institute Pilot Grant and Postdoctoral Fellowship (TRS). LBV is an investigator of the Howard Hughes Medical Institute.

## Additional information

### Funding

| Funder | Grant reference number | Author |
| --- | --- | --- |
| Howard Hughes Medical Institute | Investigator Award | Trevor R Sorrells<br>Leslie B Vosshall |
| Jane Coffin Childs Memorial Fund for Medical Research | Postdoctoral Fellowship | Trevor R Sorrells |
| Kavli Neural Systems Institute, The Rockefeller University | Postdoctoral Fellowship | Trevor R Sorrells |
| Kavli Neural Systems Institute, The Rockefeller University | Pilot Award | Trevor R Sorrells |

The funders had no role in study design, data collection and interpretation, or the decision to submit the work for publication.

### Author contributions

Trevor R Sorrells, Conceptualization, Data curation, Formal analysis, Funding acquisition, Investigation, Methodology, Supervision, T.R.S. performed all experiments and analyses in the paper with the exception of Figure 4I., Visualization, Writing – original draft, Writing – review and editing; Anjali Pandey, A.R.-V performed experiments and analyses in Figure 4I., Investigation, Methodology, Writing – review and editing; Adriana Rosas-Villegas, A.P. developed the rearing and experimental protocol for the optogenetic mosquito lines., Formal analysis, Investigation, Methodology, Visualization, Writing – review and editing; Leslie B Vosshall, Funding acquisition, Project administration, Supervision, Visualization, Writing – original draft, Writing – review and editing

### Author ORCIDs

Trevor R Sorrells http://orcid.org/0000-0002-3527-8622
Anjali Pandey http://orcid.org/0000-0001-5521-635X
Adriana Rosas-Villegas http://orcid.org/0000-0001-9114-0882
Leslie B Vosshall http://orcid.org/0000-0002-6060-8099

### Ethics

Blood feeding of mosquitoes with human volunteers was conducted according to IRB protocol LV-0652. Human volunteers gave written informed consent to participate in the experiments.
Blood feeding of mosquitoes with live anesthetized mice was conducted according to approved institutional animal care and use committee (IACUC) protocol #17108 of The Rockefeller University.

Decision letter and Author response
Decision letter https://doi.org/10.7554/eLife.76663.sa1
Author response https://doi.org/10.7554/eLife.76663.sa2

## Additional files

### Supplementary files
• Transparent reporting form

### Data availability
Data availability All data generated or analyzed during this study are included in the manuscript and Source Data Files 1-4. Large datasets are available at https://github.com/trevorsorrells/Optothermo-cycler, (copy archived swh:1:rev:9c70d0c348ad6d8a32663ca23257616f915ab06e).

The following dataset was generated:

| Author(s) | Year | Dataset title | Dataset URL | Database and Identifier |
|---|---|---|---|---|
| Sorrells TR | 2021 | A persistent behavioral state enables sustained predation of humans by mosquitoes | https://github.com/trevorsorrells/Optothermocycler | GitHub, GitHub |

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
