## [Editor Report]

This manuscript describes a female mosquito's behaviour after a brief exposure to CO2, which has long been known to trigger host-seeking in female mosquitoes. The authors develop optogenetic tools in Aedes aegypti which enables them to control the delivery of 'fictive' CO2 to mosquitoes. They use this to show that a brief pulse of fictive CO2 alters the behavioural state of female mosquitoes, which lasts about 15 minutes. The study provides new insights into how activation of CO2-sensing olfactory neurons alters the behavioural state of a mosquito towards sensory cues to increase host-seeking behaviors and will be of great value to the vector biology community, as well as to neurobiologists in general.

---

## [Decision Letter]

**Decision letter after peer review:**

Thank you for submitting your article "A persistent behavioral state enables sustained predation of humans by mosquitoes" for consideration by *eLife*. Your article has been reviewed by 3 peer reviewers, including Sonia Sen as Reviewing Editor and Reviewer #2, and the evaluation has been overseen by Claude Desplan as the Senior Editor. The following individual involved in review of your submission has agreed to reveal their identity: Marcus C Stensmyr (Reviewer #1).

Essential revisions:

We appreciate the work presented here for both its quality and the conceptional ground it breaks. We do have a few suggestions for the authors that we think they should be able to address.

1. The authors should lay out the rationale for the choice of light intensity and duration for optogenetic stimulation. Based on their response to earlier reviewers, it looks like a number of conditions were attempted before settling on the ones presented in this manuscript. It would be nice to have this information laid out here.

2. The authors should show the expression pattern of the Gal4 lines in the whole brain instead of just the antennal lobe.

3. In response to earlier reviews, the authors state that they confirmed that the blood-fed mosquitoes respond to GR4 stimulation. This should be included to demonstrate that blood-fed mosquitoes are responsive to other cues, but un-responsive to CO2 stimulation.

Aside from this essential revisions, there are a few clarifications required in the detailed reviews attached below. The authors should respond to these wherever possible.

*Reviewer #1 (Recommendations for the authors):*

All good. Paper has already been thoroughly reviewed, so no need to drag this out further.

*Reviewer #2 (Recommendations for the authors):*

We enjoyed reading this manuscript and appreciate it for the ground it breaks. We have a few comments that the authors might want to consider.

At first glance, figure 2 E-H, suggests that (1) light alone is sufficient to induce the persistent state, and (2) that this state lasts longer in the presence of light alone, than in the presence of light and heat. This initially seems counterintuitive to the case that's being made (I believe two of the earlier reviewers have raised related issues). Perhaps directly addressing these aspects of the data in the Results section, and discussing possibilities in the discussion will help these apparent discrepancies. As the authors later point out, when both heat and light are presented together, there's much more probing (more indicative of the host seeking state?). Moreover, it's possible that the mosquitoes are 'giving up' when the two host cues haven't resulted in a meal, but are still searching for the second cue when there's only CO2?

Is there a discrepancy between figure 4D-F and J? D-F suggest that CO2 can potentiate response to heat for 4 minutes (read-out: walking/probing), but J suggests that it can do so for upto 14 minutes (read-out: blood-feeding). Can the authors please address this?

It would be nice to see the expression pattern of the two Gal4 lines used in the whole brain. There seems to be some expression beyond the glomerulus of interest. While this will not negate the results presented here, it would be nice to show.

The modulation of the behavioural state in figure 3 is very nice! It would have been nice to show that in the blood fed state the mosquitoes are responsive to other unrelated cues such as those relevant for oviposition. In the response to earlier reviews, the authors mention testing Gr4 stimulation. Perhaps they could include it as a supplementary figure to make this point?

*Reviewer #3 (Recommendations for the authors):*

We are impressed by the data presented in the paper. It is an incredible achievement!

A concern is that the activities of the CO2-sensing (Gr3+) neurons upon red-light stimulation were not directly monitored. For Chrimson, as a light-gated ion channel, it is possible that light activation could over-activate the sensory neuron. As such, instead of activation, the authors might be inhibiting firing of these neurons. If so, the behavioral assays might be testing a mixture of fast activation and extended inhibition of the CO2 neurons (while assuming only activation). This occurs in *Drosophila* experiments with OR neurons expressing Chrimson: low intensity light conditions are sufficient to strongly activate olfactory neurons, whereas light intensity conditions often used for optogenetic activation of central brain neurons can lead to inhibition of the neuron for extended periods of time (presumably as the sensilla restores the ion concentrations in the lymph).

This can be addressed by SSR recordings of cpA neurons under different light conditions (ideally replicating the intensity of the light stimuli encountered during behavioral experiments). This will give a clear read-out regarding the cpA activity levels likely used during the behavioral assays. These could also be performed in conjunction with different CO2 concentration stimuli, which would have the added bonus of associating light intensity stimuli to potential CO2 concentrations.

Alternatively, the authors could present a behavioral dose response curve (assaying one of their monitored behaviors) under increasing light intensity conditions. This might demonstrate faithful host-seeking behaviors with the intensity conditions they used, but reduced/no behaviors/atypical behaviors as light intensity conditions are raised. It's possible the authors already did these experiments to titrate light conditions for their experiments. This would at least demonstrate changes to behaviors might occur at higher light conditions (hopefully not used by the authors), and act as a warning to other researchers that intensity conditions need to be carefully calculated and controlled during optogenetic experiments.

---

## [Author Response]

Essential revisions:We appreciate the work presented here for both its quality and the conceptional ground it breaks. We do have a few suggestions for the authors that we think they should be able to address.1. The authors should lay out the rationale for the choice of light intensity and duration for optogenetic stimulation. Based on their response to earlier reviewers, it looks like a number of conditions were attempted before settling on the ones presented in this manuscript. It would be nice to have this information laid out here.

We have provided more details on the choice of light intensity and duration in the revised manuscript.

2. The authors should show the expression pattern of the Gal4 lines in the whole brain instead of just the antennal lobe.

Expression patterns of *Gr3 > CsChrimson* and control strains in the whole brain are shown in new Figure 1—figure supplement 1A.

3. In response to earlier reviews, the authors state that they confirmed that the blood-fed mosquitoes respond to GR4 stimulation. This should be included to demonstrate that blood-fed mosquitoes are responsive to other cues, but un-responsive to CO2 stimulation.

Responses to host stimuli of blood-fed *Gr4 > CsChrimson* mosquitoes are now included in new Figure 3—figure supplement 1.

Aside from this essential revisions, there are a few clarifications required in the detailed reviews attached below. The authors should respond to these wherever possible.Reviewer #1 (Recommendations for the authors):All good. Paper has already been thoroughly reviewed, so no need to drag this out further.

Thank you – greatly appreciated!*Reviewer #2 (Recommendations for the authors):*

We enjoyed reading this manuscript and appreciate it for the ground it breaks. We have a few comments that the authors might want to consider.At first glance, figure 2 E-H, suggests that (1) light alone is sufficient to induce the persistent state, and (2) that this state lasts longer in the presence of light alone, than in the presence of light and heat. This initially seems counterintuitive to the case that's being made (I believe two of the earlier reviewers have raised related issues). Perhaps directly addressing these aspects of the data in the Results section, and discussing possibilities in the discussion will help these apparent discrepancies. As the authors later point out, when both heat and light are presented together, there's much more probing (more indicative of the host seeking state?). Moreover, it's possible that the mosquitoes are 'giving up' when the two host cues haven't resulted in a meal, but are still searching for the second cue when there's only CO2?

This is an astute observation and the interpretation of the result made by the reviewer is one we also considered. However, we did not include this in the publication because the effect is not statistically significant, and it was not apparent in all experiments run. Specifically, we did not detect a shorter half-life to light + heat than light alone nor did we detect a difference in total probing or walking. We did find that the major response is an additive effect on probing when the stimuli are combined, so we included this result in the publication. This idea of whether heat shortens the persistent state is an interesting area for future experiments and we thank the reviewer for the suggestion!

Is there a discrepancy between figure 4D-F and J? D-F suggest that CO2 can potentiate response to heat for 4 minutes (read-out: walking/probing), but J suggests that it can do so for upto 14 minutes (read-out: blood-feeding). Can the authors please address this?

We think that this difference arises from the signal to noise in these responses contributing to the ability to detect statistical differences. The effect in Figure 4D-F is measured at the level of the probing and walking response of individual mosquitoes which is highly variable. Figure 4D is measured as the proportion of mosquitoes that engorged on blood using 12 replicates of 14 mosquitoes, which is less variable.

It would be nice to see the expression pattern of the two Gal4 lines used in the whole brain. There seems to be some expression beyond the glomerulus of interest. While this will not negate the results presented here, it would be nice to show.

We have added the whole brain images to Figure 1 —figure supplement 1A. The expression beyond the glomerulus is derived from the *3XP3-dsRed* marker used to track the *Gr3-QF2* insertion. The *3XP3* enhancer is broadly expressed in the visual system and on occasion in central brain neurons. This is a known issue (Shankar et al., 2021, Younger et al., 2022) and is usually circumvented by using a different fluorophore to mark the strain than the one used for the experiment. In this study, both the transgene marker and the optogenetic reagent are marked with fluorescent proteins in the red spectral range. We know from prior work using a green reporter that the Gr3-QF2 is very selectively expressed in carbon dioxide-expressing neurons (Younger et al., 2022).

The modulation of the behavioural state in figure 3 is very nice! It would have been nice to show that in the blood fed state the mosquitoes are responsive to other unrelated cues such as those relevant for oviposition. In the response to earlier reviews, the authors mention testing Gr4 stimulation. Perhaps they could include it as a supplementary figure to make this point?

Thank you for the suggestion for this experiment. We have added the *Gr4* data to Figure 3—figure supplement 1. We found that there was a response in both *Gr3 > CsChrimson* and *Gr4 > CsChrimson* mosquitoes to all stimuli except for *Gr3 > CsChrimson* in response to light. We added the following text to the manuscript:

“For comparison, we asked whether blood-fed females could respond to the optogenetic sugar stimulus by activating *Gr4* sensory neurons. We found that blood-fed females had reduced responses to fictive sugar, but unlike fictive CO_2_, the response was still detectable (Figure 3—figure supplement 1B-D). This demonstrates that blood-fed females specifically show a complete loss of the persistent state elicited by fictive CO_2_.”

Reviewer #3 (Recommendations for the authors):We are impressed by the data presented in the paper. It is an incredible achievement!A concern is that the activities of the CO2-sensing (Gr3+) neurons upon red-light stimulation were not directly monitored. For Chrimson, as a light-gated ion channel, it is possible that light activation could over-activate the sensory neuron. As such, instead of activation, the authors might be inhibiting firing of these neurons. If so, the behavioral assays might be testing a mixture of fast activation and extended inhibition of the CO2 neurons (while assuming only activation). This occurs in *Drosophila* experiments with OR neurons expressing Chrimson: low intensity light conditions are sufficient to strongly activate olfactory neurons, whereas light intensity conditions often used for optogenetic activation of central brain neurons can lead to inhibition of the neuron for extended periods of time (presumably as the sensilla restores the ion concentrations in the lymph).

The reviewer raises an interesting point about the precise nature of the neural responses to light. We do not know whether there is activation followed by inhibition or just activation alone under the conditions of our experiments. Our light dose-response curve (new Figure 1—figure supplement 1B-D) suggests that the behavioral response is prolonged at low as well as higher light intensities. Furthermore, if our light intensity results in neuronal activation followed by extended inhibition as seen in high intensity activation in *Drosophila* olfactory neurons, this would suggest that ongoing activity in these neurons is not required for the persistent behavioral response to host stimuli.

This can be addressed by SSR recordings of cpA neurons under different light conditions (ideally replicating the intensity of the light stimuli encountered during behavioral experiments). This will give a clear read-out regarding the cpA activity levels likely used during the behavioral assays. These could also be performed in conjunction with different CO2 concentration stimuli, which would have the added bonus of associating light intensity stimuli to potential CO2 concentrations.

This is an excellent suggestion, and we plan to perform these experiments in future work. We feel that the conclusions in the present paper do not depend on the precise mapping of CO2 concentration to light.

Alternatively, the authors could present a behavioral dose response curve (assaying one of their monitored behaviors) under increasing light intensity conditions. This might demonstrate faithful host-seeking behaviors with the intensity conditions they used, but reduced/no behaviors/atypical behaviors as light intensity conditions are raised. It's possible the authors already did these experiments to titrate light conditions for their experiments. This would at least demonstrate changes to behaviors might occur at higher light conditions (hopefully not used by the authors), and act as a warning to other researchers that intensity conditions need to be carefully calculated and controlled during optogenetic experiments.

Thank you for this suggestion. We have added the light dose-response curve in Figure 1—figure supplement 1. The results demonstrate that the behavior responses are graded throughout the range of intensities that we delivered. We did not see atypical behaviors at the highest light intensities, but these may be lower maximum intensities than have been used in other systems. We chose intermediate intensities for the experiments throughout the paper to be cautious. The intensities we chose are similar to experiments typically performed in *Drosophila* for optogenetic activation of olfactory sensory neurons. We have added the following sentences throughout the manuscript to clarify our choice of light intensity.

“The proportion of mosquitoes responding increased with light intensity (Figure 1—figure supplement 1B-D).”

“Varying the light intensity changed the proportion of mosquitoes responding but not the duration of the response (Figure 1—figure supplement 1).”

“The light intensity chosen was an intermediate intensity as determined by a light-behavior dose-response curve (Figure 1—figure supplement 1B-D).”

“Red light stimuli were 627 nm at an intensity of 12 µW/mm^2^, chosen as an intermediate intensity that allowed the possibility of both an increase and decrease in the behavioral response.”

References

Shankar S, Tauxe GM, Spikol ED, Li M, Akbari OS, Giraldo D, McMeniman CJ. 2021. Synergistic coding of carbon dioxide and a human sweat odorant in the mosquito brain. *bioRxiv* 10.1101/2020.11.02.365916

Younger MA, Herre M, Goldman OV, Lu T-C, Caballero-Vidal G, Qi Y, Gilbert ZN, Gong Z, Morita T, Rahiel S, Ghaninia M, Ignell R, Matthews BJ, Li H, Vosshall LB. 2022. Non-canonical odor coding in the mosquito. *BioRxiv DOI_101101/20201107368720v2*